**Data Availability Statement:** All relevant data are within the manuscript.

# Inclusion of soybean and linseed oils in the diet of lactating dairy cows makes the milk fatty acid profile nutritionally healthier for the human diet

Mauricio X. S. Oliveira[1][☯], Andre S. V. Palma[1][☯], Barbara R. Reis[1][☯], Camila S. R. Franco[1][☯], Alessandra P. S. Marconi[1][☯], Fabiana A. Shiozaki[1][☯], Leriana G. Reis [ID][1][☯], Marcia S. V. Salles[2][☯], Arlindo S. Netto [ID][1][☯] *

1 Faculty of Animal Science and Food Engineering, University of São Paulo, Pirassununga, São Paulo, Brazil, 2 Animal Science Institute, Ribeirão Preto, São Paulo, Brazil

☯ These authors contributed equally to this work.
* saranetto@usp.br

## Abstract

Fluid milk and its derivatives are important dietary ingredients that contribute to daily nutrient intake of the modern Homo sapiens. To produce milk that is healthier for human consumption, the present study evaluated the effect of adding soybean oil and linseed oil in the diet of lactating cows. The fatty acid profile of milk, milk composition, and the blood parameters of cows were evaluated. Eighteen Holstein cows were distributed in a replicated Latin square design and distributed according to the following treatments: 1) Control (CC): traditional dairy cow diet, without addition of oil; 2) Soybean oil (SO): 2.5% addition of soybean oil to the traditional diet, as a source of omega-6; 3) Linseed oil (LO): 2.5% addition of linseed oil in the diet as a source of omega-3. Milk production was not affected, but oil supplementation decreased feed intake by 1.93 kg/cow/day. The milk fat percentage was significantly lower when cows were supplemented with vegetable oil (3.37, 2.75 and 2.89% for CC, SO and LO, respectively). However, both soybean and linseed oils decreased the concentration of saturated fatty acids (66.89, 56.52 and 56.60 g/100g for CC, SO and LO respectively), increased the amount of unsaturated fatty acids in milk (33.05, 43.39, and 43.35 g/100g for CC, SO and LO respectively) and decreased the ratio between saturated/unsaturated fatty acids (2.12, 1.34, and 1.36 for CC, SO and LO respectively). Furthermore, SO and LO increased significantly the concentration of monounsaturated fatty acids (29.58, 39.55 and 39.47 g/100g for CC, SO and LO respectively), though it did not significantly alter the level of polyunsaturated fatty acids in milk fat (3.57, 3.93 and 3.98 g/100g for CC, SO and LO respectively). Supplementation with LO enhanced the concentration of omega-3 fatty acids on milk (0.32, 0.36, and 1.02 for CC, SO and LO respectively). Blood variables aspartate aminotransferase, gamma glutamyl transferase, urea, albumin, creatinine and total proteins were not altered. On the other hand, total cholesterol, HDL and LDL were greater in the group supplemented with vegetable oils. Supplementation with vegetable oils reduced the dry matter intake of cows, the fat content of milk, and improved saturated/unsaturated fatty acid ratio of milk fat. Compared to the SO treatment, animals fed LO produced milk with

**Funding:** This work was funded by São Paulo Research Foundation (FAPESP) São Paulo, Brazil, research funding (FAPESP Process 2015/ 19393-8); two scholarships (FAPESP Processes 2016/ 23735-4 and 2016/23797-0), and one master's scholarship - Coordenação de Aperfeiçoamento de Pessoal de Nível Superior (CAPES), for Mauricio X. S. Oliveira. The funders had no role in study design, data collection and analysis, decision to publish, or preparation of the manuscript.

**Competing interests:** The authors have declared that no competing interests exist.

greater content of omega-3, and a more desirable omega-6/omega-3 ratio on a human nutrition perspective. Thus, the inclusion of SO and LO in the diet of lactating dairy cows makes the milk fatty acid profile nutritionally healthier for the human consumption.

## Introduction

The fatty acid (FA) profile of the human diet has changed considerably since the industrial revolution [1]. The requirements for practicality and fast meal preparation have led to increased intake of industrial foods, that are typically rich in saturated and omega-6 FAs [2]. In the last 100 to 150 years, the absolute and relative change of omega-6 and omega-3 fatty acids in the food supply of Western societies have led the Western diet to reach a ratio of 20:1 omega-6 and omega-3 [3], which is greater than the 5:1 to 10:1 recommended by the World Health Organization [4]. Some studies suggests that diets rich in saturated and omega-6 fatty acids might shift the physiological state to one that is pro-inflammatory, pro-thrombotic, and pro-aggregatory, with increase in blood viscosity, vasospasm, and vasoconstriction, which are factors associated with the development with coronary heart disease [5] and other non-infectious diseases [6]. Hence, increasing omega-3 polyunsaturated fatty acid (PUFA) consumption, as well as lowering intakes of saturated fat and omega-6 PUFA is desirable [7].

Historically, milk from domestic cows has been an important nutrient source for over 8,000 years in human societies exploiting dairy breeds. It is believed that this relation has led to a geneculture coevolution between domestic cattle and human culture [8]. Nevertheless, in the last decades, a trend to decrease in fluid milk consumption is observed in certain countries, like the United States and Canada [9–11]. The claimed negative health effects that have been attributed to milk and milk products may partly explain this trend. This criticism has arisen especially because milk fat contains a high fraction of saturated fatty acids, which is assumed to contribute to heart diseases, weight gain, and obesity [12]. Since more dietary PUFA, especially omega-3, might decrease occurrence of coronary and non-infectious disease [4,5], there has been a great deal of interest in manipulating the fatty acid profile of milk fat to respond to consumer demand [13,14].

There are two essential fatty acids in human nutrition: linoleic acid (C18:2 omega-6) and alpha-linolenic acid (C18:3 omega-3) [15]. Humans must obtain essential fatty acids from foods because the human body cannot synthesize it [15]. Linoleic acid is prevalent in corn, sunflower, and mainly in soybean oil, whereas alpha-linolenic acid is predominating in canola, rapeseed, fish oil, and in linseed oil [16]. The use of polyunsaturated oils in the ruminant diet has been shown to be an efficient alternative to enhancing the lipid composition of milk [17,18].

Enhancing the FA profile of milk is beneficial for the dairy industry [19]. Nevertheless, ruminal biohydrogenation (RB), which transforms unsaturated fatty acids (UFA) into saturated fatty acid (SFA), is a major obstacle on improving the nutrition value of milk. The RB process has long been known to occur in the rumen, the first compartment of the ruminant stomach, as the result of microbial metabolic activity [20]. However, there is a limitation on the RB of UFA, and some UFA leave the rumen without being bio-hydrogenated, which means that there might be some potential to enhance the healthiness of ruminant meats and milk by increasing their PUFA composition, particularly omega-3 FA [21].

Dietary FAs can be further processed in the ruminant mammary tissue. The enzyme stearoyl-CoA desaturase in the mammary secretory cell of the mammary gland introduces a cis-double bond exclusively at the Δ9 position of fatty acid [22]. This reaction results in conversion

of SFA to monounsaturated fatty acid (MUFA) and specific conjugated linoleic acid (CLA) isomers only found in ruminant products [23]. The predominant CLA isomer in milk fats is rumenic acid (cis-9, trans-11 C18:2), and the majority of milk CLA is derived by *de novo* synthesis in the mammary gland by Δ9 desaturation of vaccenic acid (trans-11 C18:1) arising from RB [24]. It is believed that certain CLA can contribute to human health. For instance, rumenic acid exhibits anticarcinogenic and antiatherogenic properties in animal models [25].

The objective of this study was to evaluate the effect of the inclusion of soybean oil, as a source of omega-6 FA, and linseed oil, as a source of omega-3 FA, in the diet of lactating cows on milk production, milk composition, milk fatty acid profile, and blood parameters of lactating dairy cows under tropical conditions. Our hypothesis is that supplementing dairy cows with vegetable oils will enhance the FA profile of milk by decreasing the saturated fatty acid content (e.g. C10:0, C12:0, C14:0, C16:0) and increasing the unsaturated fatty acid content (e.g. C18:2 cis9 cis 12, C18:2 cis9 trans11, and C18:3 omega-3).

## Material and methods

The Ethics Committee on Animal Experimentation at the Faculty of Animal Science and Food Engineering, University of São Paulo approved this project, under the protocol number 4939070317.

### Study design

This study was conducted at Dairy Cattle sector in the University of São Paulo, Campus Fernando Costa, in the city of Pirassununga, São Paulo, Brazil (21˚57'19.5"S and 47˚27'10.2"W). The minimum and maximum temperature recorded during the study period were respectively 15.2 and 27.7˚C.

A total of eighteen lactating Holsteins dairy cows, with an initial body weight mean of 616.33 ± 152 kg, mean body condition score of 3.25, and mean initial milk yield of 26.46 ± 6 kg/day, were selected and housed in the stall barn. Study animals were distributed on a 6 contemporary, 3x3 Latin square design, with the experimental unit being animal during each experimental period (n = 54 experimental units). The study lasted for 94 days. The first ten days were designed for adaptation of animals to the stalls, in which they were kept during the whole study. During this interval, all animals were fed the same diet. The remaining 84 days corresponded to three experimental periods consisting of 28 days each, whereas the first 21 consisted on adaptation to the new diet, and the seven remaining days were designed for data and sample collection.

The treatments consisted of the following experimental diets: 1) Control (CC): traditional dairy cow diet, with no addition of oil; 2) Soybean oil (SO): 2.5% addition of soybean oil to the traditional diet, as a source of omega-6; 3) Linseed oil (LO): 2.5% addition of linseed oil in the diet as a source of omega-3. Both of SO and LO were added in the order of 2.5% of the total dry matter, in the substitution of 2.5% of corn, as shown in Table 1. These levels were chosen because according to previous studies from our research group [13,14], they could enhance the milk FA profile without significantly altering production parameters of dairy cows, e.g. milk yield.

In the present study, soybean and linseed oil had a total of 84.2% and 88.4% of unsaturated fatty acids (UFA), respectively. Experimental diets were formulated according to the recommendations of the National Research Council for dairy cattle [26] (Holstein in lactation, body weight: 680 kg, milk production: 30 kg/day, dry matter intake: 19.4 kg/day or 2.86% of body weight). The diets consisted of 50% experimental concentrate and 50% corn silage. The experimental diets and water were provided ad libitum. The ingredients and chemical composition of the diet are given on Table 1.

**Table 1. Ingredients proportion and chemical composition of the experimental diets expressed on dry matter percentage.**

| | Treatment[1] | | |
|---|---|---|---|
| **Ingredients %** | **CC** | **SO** | **LO** |
| **Ground Corn** | **30.5** | **28.0** | **28.0** |
| Soybean Meal | 17.0 | 17.0 | 17.0 |
| Soybean Oil | - | 2.50 | - |
| Linseed Oil | - | - | 2.50 |
| Urea | 0.50 | 0.50 | 0.50 |
| Limestone | 0.30 | 0.30 | 0.30 |
| Dicalcium Phosphate | 0.15 | 0.15 | 0.15 |
| Salt | 0.50 | 0.50 | 0.50 |
| Mineral Complex[2] | 1.05 | 1.05 | 1.05 |
| Corn Silage | 50.00 | 50.00 | 50.00 |
| **Chemical Composition DM%** | | | |
| Dry Matter | 92.01 | 92.61 | 91.68 |
| Mineral Matter | 3.38 | 3.19 | 3.18 |
| Crude Protein | 16.64 | 16.15 | 16.24 |
| Neutral Detergent Fiber | 29.97 | 29.48 | 29.42 |
| Acid Detergent Fiber | 18.73 | 18.14 | 17.99 |
| Ether Extract | 3.49 | 5.67 | 5.78 |
| Lignin | 3.91 | 3.56 | 3.61 |
| Total Carbohydrate | 75.95 | 74.01 | 74.18 |
| Non-Fiber Carbohydrate | 45.99 | 44.53 | 44.76 |
| Total Digestible Nutrients | 66.77 | 69.31 | 69.46 |

[1] CC = Control; SO = Soybean Oil; LO = Linseed Oil

[2] Composition per kilogram of product: Sulfur (S) 80g, Magnesium (Mg) 20g, Potassium (K), 20g, Manganese (Mn) 1000mg, Zinc (Zn) 2500 mg, Copper (Cu) 1500 mg, Cobalt (Co) 100mg, Iodine (I) 80 mg, Selenium (Se) 20 mg, Calcium (Ca) 180g, Phosphorus (P) 90 g, Fluor (F) 300mg.

Animals were separated by electric fences to allow individual feed intake measurement. Animals were milked twice a day, at 07:00 and 15:00. Detection of clinical mastitis was performed daily through the black mug test [27], before each milking. Animals were fed individually twice a day, right after milking. The amount of feed given each day was determined by their individual dry matter intake from the previous day, and the percentage of dietary leftovers calculated to be kept between 10 and 20% of the total feed supplied, so that there was no consumption limitation.

## Feed assay

Daily samples of the feed and orts were collected and pooled over on every collection period. Feed samples were mixed and sub-sampled, and then stored on a -21°C freezer for subsequent chemical analysis. At the end of the study, samples were defrosted and dried at 65°C in a forced-air oven for 72 h and then grounded on a 1 mm screen mill (Wiley® Mill, Thomas Scientific, Swedesboro NJ, United States). Dry matter (AOAC, 950.15), ash (AOAC, 942.05), ether extract (EE; AOAC, 920.39), crude protein (CP; AOAC, 984.13), acid detergent fiber (ADF), and lignin (AOAC, 973.18) were analyzed in the feed offered and in samples of orts according to the methods described by the Official Methods of Analysis of AOAC International [28]. The neutral detergent fiber (NDF) content of feed and orts were determined by

using α-amylase and with no addition of sodium sulphite [29,30] using the Ankon® system (ANKOM Technology, Macedon NY, USA).

## Milk production and composition

Milk production corresponded to the sum of the production of the two daily milks, which were measured on an electronic flow meter (DeLaval, Tumba, Sweden) of mechanical milking machine. The results were imported through the Alpro program (DeLaval, Tumba, Sweden) and saved on Microsoft Excel (Microsoft Corporation, Redmond WA, United States) spreadsheets. At each experimental collection period, daily, for each animal, one milk sample was collected during each milking, and mixed in a proportion of 60/40 for morning and afternoon. This collection was done during three consecutive days, on 100 mL collector tubes, containing 2-bromo- 2-nitropropane-1.3-diol (0.05%, wt/vol) preservative.

Milk samples were analyzed for fat, protein, lactose, total solids by the infrared absorption process Bentley 2000 analyzer (Bentley Instruments, Inc. Chaska, Minnesota, USA) [31]; urea nitrogen by spectrophotometric enzymatic method using the ChemSpec 150 analyzer [32], and somatic cell count by flow cytometry using the Bentley 2000® and Somacount 300® apparatus [33]. The defatted dry extract (ESD) was obtained by the difference in fat content and total milk solids.

## Fatty acid and cholesterol assay

Milk samples were subjected to determination of cholesterol and fatty acids profile according to the methodology proposed by [34]. Milk samples were collected on a plastic vial with no preservative on it and then frozen on a -20°C freezer until the end of the study. At the end of the study, samples were unfrozen and extraction of fat was conducted according to [35] and methylated according to [36].

Cholesterol determination was conducted according to the methodology proposed by [37]. In short, to the previously thawed and homogenized milk (10g), 8 ml of 50% KOH solution and 12 ml of ethyl alcohol were added, and the solution was placed in a water bath at 60°C under constant agitation for 15 minutes. Then, 10 mL of distilled water was added until complete cooling. The extraction was carried out with hexane under intense agitation. After separation, a sample of 4 ml of the hexane phase was removed, dried over $N_2$, and mixed with 0.5 ml of isopropanol until complete solubilization. To quantify cholesterol using the enzymatic methodology, the Laborlab® kit was used, composed of two color reagents, in addition to the enzymatic reactive. After the procedure specified by the kit, an absorbance reading was performed against the blank, also prepared at 499 nm. The standard curve was constituted from a standard cholesterol solution (1.006 mg/100mL), with concentration ranging from 0.01 to 0.08 mg / mL.

The FA profile of milk, soybean oil, linseed oil, and total diets were determined by gas chromatography (ThermoFinnigan®, model Trace 2000) using a fused silica capillary column, SP-2560 (100mm x 0.25 mm x 0.2 mm, supelco) and flame ionization detector (FID). Hydrogen was used as carrier gas at a flow rate of 1.8 mL/min. The oven temperature program was an initial 70°C with a holding time of 27 minutes, 215°C (4°C/minute) with a holding time of 9 minutes, and finally, an increase by 7°C/minute to 230°C, standing for five minutes for a total of 65 minutes. Vaporizer and detector temperature were 250°C and 300°C, respectively.

Certified standard butter fat (CRM-164, Commission of the European Communities, Community Bureau of Reference, Brussels, Belgium) was used to determine the recovery of FA and to calculate correction factors. Concentrations of fatty acids are expressed as g/100 g of the total fatty acid concentration. The FA profile of soybean oil, linseed oil, and silage, as well as total SO and LO is shown on Table 2.

**Table 2. Fatty acid composition (g/100 g total fatty acids) of experimental diets, soybean oil, linseed oil, and silage.**

| Fatty Acid | Treatment[1] | | | Soybean oil | Linseed oil | Silage |
|---|---|---|---|---|---|---|
| | CC | SO | LO | | | |
| C6:0 | 0.08 | 0.02 | 0.02 | NI[2] | NI | <0.01 |
| C8:0 | 0.02 | <0.01 | <0.01 | NI | NI | 0.07 |
| C10:0 | NI | <0.01 | <0.01 | NI | NI | 0.08 |
| C12:0 | 0.01 | <0.01 | <0.01 | <0.01 | <0.01 | 0.43 |
| C12:1 | NI | NI | NI | NI | NI | NI |
| C13:0 | NI | NI | NI | NI | NI | <0.01 |
| C13 anteiso | NI | NI | NI | NI | NI | NI |
| C14:0 | 0.08 | 0.09 | 0.00 | 0.08 | 0.05 | 0.39 |
| C15:0 | 0.03 | 0.02 | 0.03 | 0.01 | 0.02 | 0.10 |
| C15 anteiso | NI | NI | NI | NI | NI | 0.05 |
| C15 iso | NI | NI | NI | NI | NI | 0.05 |
| C16:0 | 16.09 | 13.6 | 10.58 | 11.13 | 5.97 | 19.56 |
| C16 iso | 0.01 | <0.01 | NI | 0.01 | <0.01 | 0.03 |
| C16:1 c-9 | 0.12 | 0.10 | 0.10 | 0.05 | 0.05 | 0.25 |
| C17:0 | 0.10 | 0.09 | 0.08 | 0.06 | 0.04 | 0.25 |
| C17 iso | NI | NI | NI | NI | NI | 0.01 |
| C17:1 | 0.03 | NI | NI | 0.04 | 0.04 | 0.04 |
| C18:0 | 2.92 | 3.33 | 3.95 | 3.56 | 5.05 | 3.13 |
| C18:1 cis-9 | 36.20 | 27.47 | 26.6 | 23.18 | 20.34 | 20.24 |
| C18:1 cis-11 | 1.20 | 1.40 | 1.08 | 1.69 | 0.89 | 1.06 |
| C18:1 cis-12 | 0.11 | 0.07 | 0.09 | 0.10 | 0.09 | 0.10 |
| C18:1 cis-13 | 0.01 | 0.03 | 0.02 | 0.05 | 0.02 | 0.02 |
| C18:2 cis-9 cis-12 | 39.67 | 47.19 | 26.68 | 52.91 | 13.62 | 43.27 |
| C18:3 n-3 | 0.21 | 0.03 | 29.85 | 5.92 | 53.38 | 0.29 |
| C18:3 n-6 | NI | <0.01 | NI | 0.02 | NI | NI |
| C20:0 | 0.49 | 0.33 | 0.25 | 0.31 | 0.14 | 0.79 |
| C20:1 | 1.93 | 5.65 | NI | 0.26 | NI | 6.34 |
| C21:0 | NI | NI | NI | NI | NI | 0.02 |
| C22:0 | 0.10 | 0.16 | 0.10 | 0.36 | 0.13 | 0.26 |
| C23:0 | NI | 0.04 | 0.03 | 0.04 | 0.02 | 0.12 |
| C24:0 | 0.30 | 0.18 | 0.19 | 0.13 | 0.09 | 0.70 |
| C24:1 | NI | NI | <0.01 | NI | NI | 0.01 |

[1] CC = Control; SO = Soybean Oil; LO = Linseed Oil

[2] NI = Not identified.

The nutritional quality of the lipid fraction was assessed using the atherogenicity index:

$$AI = (C12:0 + (4 \times C14:0) + C16:0)/ (\Sigma MUFAs) + \Sigma\omega6 + \Sigma\omega3)$$

and thrombogenicity index [38]:

$$TI = (C14:0 + C16:0 + C18:0)/[(\Sigma MUFAs \times 0.5) + (0.5 \times \Sigma\omega6) + (3 \times \Sigma\omega3) + (\Sigma\omega3/\Sigma\omega6)]$$

and the ratio of hypocholesterolemic to hypercholesterolemic fatty acids [39]

$$h/H = [(C18:1 cis-9 + C18:2 \omega6 + C18:3 \omega3 + C20:5 \omega3 + C22:6 \omega3)/(C14:0 + 16:0)]$$

## Blood metabolites

Approximately 30 ml of blood were collected at the last day of each experimental period, via puncture in the coccygeal vein. The samples were collected into two 10 ml tubes with no antico-agulant, and one 10 ml tube with fluoride. After collection, the samples were centrifuged at 2,000 g for 15 minutes at a temperature of 4˚C to obtain the plasma, which was packed into 2 ml tubes of the eppendorf type and frozen at -20˚C freezer. Triglycerides, urea, urea nitrogen, glucose, albumin, total protein, total cholesterol, HDL, LDL, creatine, gamma glutamyl transfer-ase (GGT), aspartate aminotransferase (AST) were determined by commercial kits (Randox®, Crumlin, United Kingdom) using colorimetric methods enzymatic or enzymatic kinetics. The reading was performed in an automatic blood biochemistry analyzer (SBA-200—CEL® Auto-matic Biochemistry System) and in microplate reader (Asys Mark, Model Expert Plus—UV).

## Statistical analysis

The results found in each experimental period were converted into the same Microsoft Excel file, imported, and analyzed with SAS (Statistical Analysis System, version 9.0, Cary, North Carolina, United States) [40]. Data were evaluated for residue normality and homogeneity of variances through Proc-Univariate and Proc-Glm. Somatic cell count needed to be trans-formed into log (10) (SCC-1) to meet the statistical premise of residue normality. After check-ing the normality of the data, main effects of the treatments were analyzed by the Proc-Mixed command of SAS, according to the following model:

$$Yijkl = \mu + Ti + Qj + A(Q)k + Pl + eijkl$$

Where:

Yijkl = is the observed value;

μ = general mean;

Ti = fixed treatment effect i, i = 1 to 3;

Qj = fixed effect of the Latin square j, j = 1 to 6

A (Q) k = random effect of animal k within each Latin square, k = 1 to 18;

Pl = fixed effect of period l, l = 1 to 3;

eijkl = random error associated with each observation.

Significance was declared when treatment *p*-value < 0.05. When treatments were signifi-cantly different, further investigation was conducted with the aid of the CONTRAST function of Proc-Mixed. The three treatment averages were then compared through two orthogonal lin-ear contrasts, where mean values of the CC treatment were compared to the average of SO and LO on contrast one (C1), while contrast two (C2) compared the average SO treatment with the LO treatment.

## Results

### Milk fatty acid profile and cholesterol

The effect of diets on the FA profile of milk is shown in Table 3. When compared to the CC treatment, supplementation of SO and LO decreased the content of the following FAs: C6:0, C8:00, C10:0, C12:0, C14:0, and C16:0 (*p* < 0.01). Similarly, supplementation with both types of vegetable oil decreased the content of odd- and branched chain FAs like C12:1, C13:0, C13:0

**Table 3. Effect of vegetable oil on the fatty acid profile of dairy cows' milk (g/100 g total fatty acids).**

| Fatty Acid[1] | Treatment[2] | | | SEM[3] | p-value | C1[4] | C2[5] |
|---|---|---|---|---|---|---|---|
| | CC | SO | LO | | | | |
| C4:0 | 2.87 | 2.23 | 2.49 | 0.28 | <0.01 | <0.01 | 0.14 |
| C6:0 | 1.84 | 1.07 | 1.28 | 0.08 | <0.01 | <0.01 | 0.09 |
| C8:0 | 1.13 | 0.58 | 0.72 | 0.05 | <0.01 | <0.01 | 0.05 |
| C10:0 | 2.67 | 1.32 | 1.60 | 0.11 | <0.01 | <0.01 | 0.07 |
| C10:1 | 0.30 | 0.14 | 0.17 | 0.01 | <0.01 | <0.01 | 0.07 |
| C11:0 | 0.09 | 0.29 | 0.04 | 0.01 | <0.01 | <0.01 | 0.35 |
| C12:0 | 3.46 | 1.98 | 2.22 | 0.11 | <0.01 | <0.01 | 0.14 |
| C12:1 | 0.10 | 0.05 | 0.05 | 0.09 | <0.01 | <0.01 | 0.45 |
| C13:0 | 0.15 | 0.08 | 0.09 | 0.01 | <0.01 | <0.01 | 0.39 |
| C13:0 iso | 0.04 | 0.04 | 0.04 | 0.01 | 0.90 | 0.73 | 0.77 |
| C13:0 anteiso | 0.08 | 0.05 | 0.05 | 0.01 | <0.01 | <0.01 | 0.55 |
| C14:0 | 10.36 | 7.59 | 7.98 | 0.62 | <0.01 | <0.01 | 0.25 |
| C14:0 iso | 0.09 | 0.07 | 0.08 | 0.01 | 0.07 | 0.02 | 0.75 |
| C14:1ω9 | 0.93 | 0.82 | 0.76 | 0.20 | 0.28 | 0.14 | 0.58 |
| C15:0 | 1.26 | 0.84 | 0.88 | 0.05 | <0.01 | <0.01 | 0.56 |
| C15:0 iso | 0.22 | 0.18 | 0.18 | 0.02 | 0.01 | <0.01 | 0.87 |
| C15:0 anteiso | 0.59 | 0.47 | 0.49 | 0.05 | <0.01 | <0.01 | 0.47 |
| C16:0 | 29.82 | 24.63 | 22.67 | 2.23 | <0.01 | <0.01 | 0.21 |
| C16:0 iso | 0.26 | 0.26 | 0.25 | 0.05 | 0.83 | 0.89 | 0.56 |
| C17:0 | 0.46 | 0.40 | 0.38 | 0.04 | 0.02 | 0.01 | 0.69 |
| C16:1c9 | 1.86 | 1.77 | 1.52 | 0.19 | 0.09 | 0.12 | 0.12 |
| C17:0 iso | 0.39 | 0.34 | 0.37 | 0.01 | 0.03 | 0.02 | 0.14 |
| C17:1 | 0.21 | 0.19 | 0.18 | 0.01 | 0.23 | 0.11 | 0.52 |
| C18:0 | 11.28 | 14.50 | 14.94 | 1.18 | <0.01 | <0.01 | 0.47 |
| C18:1 trans | 3.56 | 6.80 | 5.29 | 0.04 | <0.01 | <0.01 | 0.01 |
| C18:1 c9 | 20.56 | 27.18 | 27.97 | 0.91 | <0.01 | <0.01 | 0.55 |
| C18:1 c11 | 0.90 | 1.04 | 0.99 | 0.06 | 0.22 | 0.10 | 0.55 |
| C18:1 c12 | 0.38 | 0.52 | 0.52 | 0.02 | <0.01 | <0.01 | 0.96 |
| C18:1 c13 | 0.09 | 0.15 | 0.16 | 0.01 | <0.01 | <0.01 | 0.99 |
| C18:1 t16 | 0.23 | 0.36 | 0.77 | 0.02 | <0.01 | <0.01 | <0.01 |
| C18:1 c15 | 0.04 | 0.10 | 0.65 | 0.02 | <0.01 | <0.01 | <0.01 |
| C18:2 c9c12 ω6 | 2.13 | 2.62 | 2.03 | 0.12 | <0.01 | 0.19 | <0.01 |
| C18:3 ω6 | 0.02 | 0.01 | 0.01 | <0.01 | <0.01 | <0.01 | <0.01 |
| C18:3 ω3 | 0.31 | 0.35 | 0.99 | 0.03 | <0.01 | <0.01 | <0.01 |
| C18:2 c9t11 | 0.74 | 0.65 | 0.68 | 0.08 | 0.42 | 0.21 | 0.68 |
| C18:2 t10c12 | <0.01 | 0.03 | 0.01 | <0.01 | <0.01 | <0.01 | <0.01 |
| C20:0 | 0.14 | 0.15 | 0.14 | 0.02 | 0.25 | 0.37 | 0.15 |
| C20:1 | 0.03 | 0.06 | 0.06 | <0.01 | <0.01 | <0.01 | 0.25 |
| C20:2 | 0.01 | 0.01 | 0.01 | <0.01 | 0.25 | 0.17 | 0.38 |
| C20:3 ω6 | 0.14 | 0.11 | 0.09 | 0.01 | <0.01 | <0.01 | <0.01 |
| C20:3 ω3 | <0.01 | <0.01 | <0.01 | <0.01 | <0.01 | <0.01 | <0.01 |
| C20:4 ω6 | 0.15 | 0.08 | 0.08 | <0.01 | <0.01 | <0.01 | 0.55 |
| C20:5 ω3 | 0.03 | 0.02 | 0.04 | <0.01 | <0.01 | <0.01 | <0.01 |
| C21:0 | 0.01 | <0.01 | 0.03 | <0.01 | 3.67 | <0.01 | <0.01 |
| C22:0 | 0.03 | 0.03 | 0.03 | <0.01 | 0.50 | 0.43 | 0.37 |
| C22:1ω9 | <0.01 | <0.01 | <0.01 | <0.01 | 0.63 | 0.86 | 0.44 |

(*Continued*)

**Table 3.** (Continued)

| Fatty Acid[1] | Treatment[2] | | | SEM[3] | *p*-value | C1[4] | C2[5] |
|---|---|---|---|---|---|---|---|
| | CC | SO | LO | | | | |
| C22:5 | 0.04 | <0.01 | 0.03 | <0.01 | <0.01 | <0.01 | 0.26 |
| C22:2 | <0.01 | 0.00 | 0.00 | <0.01 | 0.40 | 0.18 | 0.98 |
| C22:6 ω3 | <0.01 | <0.01 | <0.01 | <0.01 | <0.01 | <0.01 | 0.71 |
| C23:0 | <0.01 | <0.01 | <0.01 | <0.01 | <0.01 | 0.01 | 0.04 |
| C24:0 | 0.03 | 0.01 | 0.01 | <0.01 | <0.01 | 0.09 | <0.01 |
| C24:1 | 0.02 | 0.01 | 0.01 | <0.01 | 0.04 | 0.01 | 0.07 |

[1] c = cis; t = trans

[2] CC = Control; SO = Soybean Oil; LO = Linseed Oil

[3] SEM = Standard Error of the Mean

[4] C1 = Contrast 1 (CC vs. SO + LO)

[5] Contrast 2 (SO vs. LO).

anteiso, C15:0, C15:0 iso, and C15:0 anteiso. Furthermore, compared to SO, milk from LO fed cows had greater concentration of C6:0, C8:0, and C10:0 FAs ($p = 0.02$, $p < 0.01$, and $p = 0.03$, respectively). Overall, supplementation of SO and LO decreased the content of short and median chain FA.

The concentration of FAs C18:0, C18:1 cis-9, C18:1 cis-12 and C18:1 cis-13 increased when cows were supplemented with either SO or LO. Compared to the CC, the content of C18:1 cis-9 was 32.2 and 36% greater in the SO and SO, respectively ($p = 0.01$). The content of C18:1 cis-9 did differ between the SO and LO treatment ($p = 0.55$). When compared to LO, the concentration of C18:2 cis-9 cis-12 increased when cows were supplemented with SO. The concentration of C18:3 omega-3 in the LO treatment were 3.2 times greater compared to the CC ($p < 0.01$) and 2.18 greater when compared to SO ($p < 0.01$). Surprisingly, supplementing cows with SO or LO did not increase the content of C18:2 cis-9, trans-11 in milk ($p = 0.21$).

Supplementation with both SO and LO decreased the percentage of saturated fatty acids and increased the percentage of unsaturated fatty acids in the milk of dairy cows; thereby the ratio between saturated/unsaturated FA was decreased (Table 4). In addition, SO and LO significantly increased the content of MUFA ($p < 0.01$) and tended to increase PUFA ($p = 0.07$). The omega-3 family fatty acids content was significantly greater when cows received LO than SO treatment. Lower omega-6/omega-3 ratio was observed when animals were fed LO. Supplementation of SO and LO significantly decreased the atherogenicity and thrombogenicity indexes of milk fat ($p < 0.01$). Moreover, the ratio of hypocholesterolemic to hypercholesterolemic fatty acids increased ($p < 0.01$). Compared to the SO treatment, milk cholesterol content from LO fed animals tended to decrease ($p = 0.09$).

## Milk production and composition

The addition of soybean oil and linseed decreased the dry matter intake in lactating cows by 1.93 kg/day ($p = 0.02$). However, milk production and fat-corrected milk production (FCM) were not affected by treatment ($p = 0.85$ and $p = 0.19$, respectively) (Table 5).

The percentage of fat and the fat yield were reduced with the inclusion SO and LO in the diet of the cows in comparison to the CC ($p < 0.01$), and there were no differences between the two oils. Similarly, compared to the CC, we observed a reduction in the percentage of total solids in the milk of cows supplemented with SO and LO ($p < 0.01$). However, this difference was not observed in total solid production in kg/day ($p = 0.46$).

**Table 4. Effect of vegetable oil on the quality of the lipid fraction of dairy cows' milk.**

| Item | Treatment[1] | | | SEM[2] | p-value | C1[3] | C2[4] |
|---|---|---|---|---|---|---|---|
| | CC | SO | LO | | | | |
| ΣCLnA, CLA | 3.1656 | 3.6131 | 3.6756 | 0.1828 | 0.0747 | 0.0247 | 0.7923 |
| ΣSFA | 66.8867 | 56.5243 | 56.6049 | 1.4408 | <0.01 | <0.01 | 0.9691 |
| ΣUFA | 33.0507 | 43.3936 | 43.3463 | 1.4379 | <0.01 | <0.01 | 0.9857 |
| ΣMUFA | 29.5831 | 39.5512 | 39.4701 | 1.3009 | <0.01 | <0.01 | 0.9656 |
| ΣPUFA | 3.5699 | 3.9347 | 3.9786 | 0.2747 | 0.1993 | 0.0761 | 0.8605 |
| Σω3 | 0.3244 | 0.3604 | 1.0223 | 0.0292 | <0.01 | <0.01 | <0.01 |
| Σω6 | 2.4832 | 2.8805 | 2.2519 | 0.208 | 0.0039 | 0.5887 | 0.001 |
| SFA/ UFA | 2.1215 | 1.3412 | 1.3644 | 0.107 | <0.01 | <0.01 | <0.01 |
| ω6/ ω3 | 7.9187 | 8.263 | 2.7245 | 0.5045 | <0.01 | <0.01 | <0.01 |
| AI | 5.5693 | 4.7929 | 4.8439 | 0.0883 | <0.01 | <0.01 | 0.6899 |
| TI | 3.1614 | 2.1552 | 1.9669 | 0.1696 | <0.01 | <0.01 | 0.4445 |
| h/ H | 0.5969 | 0.9267 | 1.0079 | 0.0859 | <0.01 | 0.01 | 0.2362 |
| Cholesterol | 9.9422 | 10.0527 | 8.7411 | 1.6494 | 0.1738 | 0.4029 | 0.0973 |

ΣCLnA (g/100g), CLA = CLnA (18:3 ω6 + 18:3 ω3) + CLA (18:2 ω6 + 9c,11t-18:2); ΣSFA(g/100g) = Σ saturated fatty acids; ΣUFA (g/100g) = Σ unsaturated fatty acids; ΣMUFA (g/100g) = Σ monounsaturated fatty acids; ΣPUFA (g/100g) = Σ polyunsaturated fatty acids; Σ ω3 (g/100g) = Σ omega-3 fatty acids; Σ ω6 (g/100g) = Σ omega-6 fatty acids; SFA/UFA = saturated/unsaturated; ω6/ ω3 = Σomega-6/Σomega-3; AI = atherogenicity index; TI = thrombogenicity index; h/H = ratio of hypocholesterolemic to hypercholesterolemic fatty acids; Cholesterol values expressed on mg/100mL.

[1] CC = Control; SO = Soybean Oil; LO = Linseed Oil

[2] SEM = Standard Error of the Mean

[3] C1 = Contrast 1 (CC vs. SO + LO)

[4] Contrast 2 (SO vs. LO).

Supplementation of SO and LO did not influence ($p \geq 0.05$) the content or the production of the following components on milk: protein, lactose, and solids non-fat. Similarly, no differences were observed in SCC or the concentration of milk MUN.

## Blood metabolites

The effect of supplementation with SO and LO on the metabolic profile of dairy cows is show on Table 6. Compared to the CC treatment, both SO and LO diets significantly increased the cholesterol levels on the plasma of dairy cows ($p < 0.01$). No statistical differences on cholesterol levels were observed between the SO and LO-fed cows ($p = 0.56$). Similarly, SO and LO diets increased HDL and LDL concentration on the plasma when compared to the CC treatment ($p < 0.01$). Still, no significant differences on HDL and LDL were observed when comparing the SO and LO diets ($p = 0.49$ and $p = 0.78$, respectively). Compared to the CC treatment, SO and LO treatments did not affect ($p \geq 0.05$) the following blood variables: AST, GGT, urea, albumin, creatinine, total proteins, CK, triglycerides, VLDL, and glucose.

## Discussion

Dietary long-chain UFAs have the potential to be incorporated into milk fat, consequently altering the profile of short and long-chain FA and the degree of saturation of milk fat [11,12]. Indeed, the ability to enhance the FA profile of milk fat is limited not by the synthesis capacity of the mammary gland, but rather by effectively having UFA escaping from biohydrogenation in the rumen [41,42]. The FA composition of certain oils, like linseed, is considered desirable

**Table 5. Effect of vegetable oil supplementation on milk composition and production on dairy cows.**

| Variable | Treatment[1] | | | SEM[2] | p- value | C1[3] | C2[4] |
|---|---|---|---|---|---|---|---|
| | CC | SO | LO | | | | |
| Milk yield (kg) | 25.87 | 26 | 25.3 | 1.57 | 0.8563 | 0.8426 | 0.6356 |
| FCM[5] | 22.92 | 21.26 | 20.96 | 1.02 | 0.1916 | 0.0725 | 0.8765 |
| Fat (%) | 3.37 | 2.75 | 2.89 | 0.2 | <0.01 | <0.01 | 0.3323 |
| Fat kg/day | 0.81 | 0.67 | 0.68 | 0.43 | 0.01 | <0.01 | 0.7367 |
| Protein (%) | 3.29 | 3.35 | 3.3 | 0.09 | 0.3875 | 0.4157 | 0.2429 |
| Protein kg/day | 0.83 | 0.84 | 0.8 | 0.04 | 0.6503 | 0.8118 | 0.384 |
| Lactose (%) | 4.35 | 4.31 | 4.37 | 0.06 | 0.0616 | 0.4574 | 0.0282 |
| Lactose kg/day | 1.14 | 1.13 | 1.12 | 0.06 | 0.8943 | 0.6515 | 0.8702 |
| TS[6] (%) | 12.01 | 11.25 | 11.61 | 0.3 | <0.01 | <0.01 | 0.4741 |
| TS kg/day | 3.04 | 2.92 | 2.86 | 0.14 | 0.4673 | 0.2464 | 0.7284 |
| SNF[7] (%) | 8.64 | 8.75 | 8.71 | 0.13 | 0.1152 | 0.0489 | 0.5262 |
| SNF kg/day | 2.23 | 2.24 | 2.18 | 0.11 | 0.8292 | 0.9122 | 0.5456 |
| SCC[8] (cel/mL) | 247.15 | 231.33 | 266.09 | 62.27 | 0.5244 | 0.9578 | 0.2601 |
| MUN[9](mg/dL) | 14.39 | 12.23 | 12.64 | 0.69 | 0.5282 | <0.01 | 0.3864 |

[1] CC = Control; SO = Soybean Oil; LO = Linseed Oil

[2] SEM = Standard Error of the Mean

[3] C1 = Contrast 1 (CC vs. SO + LO)

[4] Contrast 2 (SO vs. LO)

[5] FCM = 4.0% fat-corrected milk

[6] TS = total solids

[7] SNF = solids non-fat

[8] SCC = Somatic Cell Count

[9] MUN = Milk Urea Nitrogen.

from a human health perspective and thus their inclusion in the diet of dairy cattle might represent a mean of achieving a more desirable FA profile in milk fat.

## Milk fatty acid profile and cholesterol

Inclusion of SO and LO led to a significant decrease on the content of the following saturated FAs: C4:0, C6:0, C8:00, C10:0, C12:0, C14:0, and C16:0.

According to [43], the short-chain fatty acids (4 to 8 carbons) and medium-chain fatty acids (10 to 14 carbons) content in milk arise almost exclusively from *de novo* synthesis of FA in the epithelial cells of the mammary gland. On the other hand, long-chain fatty acids (>16 carbons) are derived from the uptake of circulating lipids, and fatty acids of 16 carbons in length originate from both sources [43]. It is suspected that supplementation of UFA in the diet of dairy cows can reduce milk fat content because of the production of certain trans FAs, by incomplete ruminal biohydrogenation of dietary UFA [44]. These trans FA are absorbed into the gut and reach the mammary gland. It is belief that once these trans FAs reach the mammary gland, the expression of lipogenic enzymes (e. g., acetyl-CoA carboxylase and fatty acid synthase) that act in *de novo* synthesis of fatty acids is reduced [41]. Indeed, in the present study, SO and LO supplementation reduced FA concentration up to C17:0 and increased the concentration of C18:2 trans-10 cis-12, which is one of the trans FAs mainly responsible for the fat reduction in milk [45]. The presence of this FA in milk might indicate a lower lipogenic enzyme activity in *de novo* synthesis in the epithelial cells of the mammary gland. Other authors [13] have reported similar results. The decrease in C16:0 FA in milk fat, observed with

**Table 6. Effect of vegetable oil on blood parameters of dairy cows.**

| Variable | Treatment[1] | | | SEM[2] | p-value | c1[3] | c2[4] |
|---|---|---|---|---|---|---|---|
| | CC | SO | LO | | | | |
| AST U/L[5] | 72.94 | 74.88 | 68.28 | 2.82 | 0.29 | 0.65 | 0.14 |
| GGT U/L[6] | 33.5 | 34.75 | 33.72 | 2.27 | 0.92 | 0.79 | 0.75 |
| Urea mg/dL | 41 | 38.6 | 40.17 | 1.53 | 0.55 | 0.40 | 0.48 |
| Albumin g/dL | 3.51 | 3.49 | 3.51 | 0.02 | 0.90 | 0.78 | 0.71 |
| Creatinine mg/dL | 1.01 | 0.99 | 1.02 | 0.04 | 0.72 | 0.97 | 0.42 |
| Total Proteins g/dL | 7.4 | 7.29 | 7.31 | 0.01 | 0.71 | 0.42 | 0.85 |
| CK U/L[7] | 111.61 | 139.99 | 94.72 | 17.13 | 0.29 | 0.90 | 0.12 |
| Cholesterol mg/dL | 149.39 | 202.15 | 194.39 | 9.27 | <0.01 | <0.01 | 0.56 |
| Triglycerides mg/dL | 12.5 | 14.57 | 14.28 | 0.84 | 0.19 | 0.07 | 0.81 |
| HDL[8] mg/dL | 73.78 | 95.26 | 90.67 | 8.69 | <0.01 | <0.01 | 0.50 |
| LDL[9] mg/dL | 68.5 | 96.22 | 99.24 | 7.66 | 0.01 | <0.01 | 0.78 |
| VLDL[10] mg/dL | 2.44 | 2.83 | 2.83 | 0.18 | 0.24 | 0.09 | 0.98 |
| Glucose mg/ dL | 61.99 | 61.18 | 62.18 | 1.34 | 0.55 | 0.99 | 0.28 |

[1]CC = Control; SO = Soybean Oil; LO = Linseed Oil

[2]SEM = Standard Error of the Mean

[3]C1 = Contrast 1 (CC vs. SO + LO)

[4]Contrast 2 (SO vs. LO)

[5]AST = aspartate aminotransferase

[6]GGT = gamma glutamyl transferase

[7]CK = creatinine kinase

[8]HDL = high-density lipoprotein

[9]LDL = Low-density lipoprotein

[10]VLDL = Very low-density lipoprotein.

vegetal oil supplementation, might be of interest on a human nutrition perspective because palmitic acid appears to be related to increased blood cholesterol [43]. Cholesterol, in turn, is a complex substance that has many functions in the body [46]. However, an increase and accumulation in the organism can lead to increased concentration in the blood, which can eventually lead to the development of coronary diseases, like atherosclerosis, arterial hypertension, diabetes mellitus and formation of gallstones [47]. The concentration of C16 on milk fat did not differ between the SO and LO treatments.

Similarly to even chain FA with a carbonic chain from C4:0 to C16:0, supplementation with SO and LO also decreased the concentration of odd-chain FA in milk fat. According to [48], ruminal microbes are the main producers of the odd chain FA found in the ruminant tissue and inclusion of PUFA in the diet can alter the rumen environment, which in turn alters microbial fermentation and FA production in the rumen [49]. Supplementation of cis-9, cis-12 18:2 have been shown to affect cellulolytic bacteria [50], and cellulolytic bacteria synthesize iso FA. On the other hand, amylolytic bacteria produce elevated levels of anteiso and linear odd-chain FA while originating relatively low levels of iso FA [51]. We believe that the PUFA supplementation in the present study might have affected the two bacteria population, as both iso and antesio FAs concentration in milk were lower when animals were fed SO and LO. Similarly to our results, other studies where cows were fed soybean oil [52] and linseed oil [53] also found a significant decrease in the concentration of odd-chain fatty acids in milk.

The significant increase of C18:0 concentration on milk observed when cows were fed SO and LO indicates completion of the RB process to a certain extent, given that C18:0 is the final

step for complete biohydrogenation of C18:2 cis-9 cis-12 and C18:3 omega-3. On the other hand, many of these fatty acids must pass through the rumen without suffering biohydrogenation or suffering partial biohydrogenation of fatty acids, which is indicated by the significant increase in intermediate FA of RB as trans FA, e.g., C18:2 trans-10 cis-12. On a human nutrition perspective, evidence suggest that stearic acid does not increase serum cholesterol concentration, and is not atherogenic [54,55]. The greater amount of C18:1 cis-9 and its isomers found in the milk of dairy cows fed SO and LO is in agreement with previous research [53,56]. The increase of oleic acid can be the result of partial biohydrogenation of C18:2 and C18:3 FA rumen [17]. Diets with great amounts of monounsaturated FA, like C18:1, have been shown to reduce plasma cholesterol, LDL-cholesterol, and triacylglycerol concentrations on human blood [57]. Moreover, replacement of SFA to cis-unsaturated fatty acids, like 18:1, might reduce risk for coronary artery disease on humans [55].

In the present study, the concentration of PUFA in milk tended to increase when cows were fed SO and LO. Linoleic acid (18:2 omega-6) was greater on milk from SO-fed cows, and alpha-linolenic acid (C18:3 omega-3) concentration was greater on milk from LO-fed cows. Once these PUFA pass the rumen, they are absorbed through the intestinal epithelium. Inside the enterocytes, these PUFAs are again re-esterified into triglycerides and transported via chylomicrons through the lymphatic system and flow to the thoracic duct where they enter the blood system [58]. Unlike other nutrients absorbed from the gastro intestinal tract, dietary FAs enter the general circulation directly and are used by all body tissues, like the mammary gland, without a preliminary processing by the liver. When these PUFAs reach the mammary gland, they will increase the unsaturated fatty acids in milk fat [58]. On a human nutrition perspective, these fatty acids can be beneficial as they can be converted to fatty acids with 20 carbon atoms, like arachidonic acid (C20:4 omega-6) and eicosapentaenoic acid (C20:5 omega-3) [59]. Nevertheless, arachidonic acid, derived from linolenic acid, may enhance blood platelet aggregation; hence, it can increase the coronary risk [60]. Arachidonic acid has the potential to partially block the conversion of omega-6 FAs to harmful eicosanoids. Hence, it can reduce cardiovascular risk and inhibit tumor genesis [61].

The intake ratio of omega-6/omega-3 FA in the human diet was modified throughout the evolution of the species. The Mesolithic man is estimated to have had a ratio of 1–4:1 between the omega-6 and omega-3 FA, whereas most societies nowadays have this ratio varying between 10–14:1 [62]. Indeed, the search for a healthier diet has been the aim of several studies on human health because of numerous cases of non-communicable diseases that trigger the death of thousands of people worldwide [63]. In this perspective, the need for balancing the omega-6/omega-3 ratio is beneficial. Evidence shows that excessive omega-6 intake might lead to pro-inflammatory effects, increasing the production of cytokines with vasoconstriction properties, which promotes platelet aggregation [1]. Indeed, platelet aggregation is related to occurrence of cardiovascular and inflammatory diseases, like arthritis, asthma, and ulcerative colitis [3]. Omega-3 FA, on the other hand, play an essential role in preventing cardiovascular diseases because it produces eicosanoids with less inflammatory power and even anti-inflammatory properties [61]. These eicosanoids promotes vasodilation and inhibits platelet aggregation, which are functions that are also related to the prevention of hypertension, atherosclerosis, hypercholesterolemia, arthritis and other autoimmune and inflammatory diseases as well as various cancers [7]. In the present study, milk of cows fed LO can be a valid option in the search for healthier food items that has the potential to prevent certain coronary diseases. Milk from cows fed LO resulted in an omega-6/omega-3 ratio of 2.7:1, whereas milk from CC-fed cows 7.9:1 and SO was 8.3:1.

Supplementation with SO and LO significantly decreased the atherogenicity and thrombogenicity indexes in the milk of dairy cows. The atherogenicity is the ability to induce

atherosclerosis formation, whereas thrombogenicity means the ability to promote heart attacks and strokes [38]. Lower values for both indices are desirable, as the lower the amount of atherogenic fatty acids present in a given fat or oil, the greater the potential for prevention against the development of coronary heart disease [38]. The ratio of hypocholesterolemic to hypercholesterolemic fatty acids index works the opposite way of the previous indexes, the higher the hypocholesterolemic to hypercholesterolemic fatty acids index is, the better nutrition is, and the lower the risk of cardiovascular disease is [39]. There is no recommended value for none of these indices in dairy products. Nevertheless, it is considered that the lower the value of atherogenicity and thrombogenicity indices, and a greater hypocholesterolemic to hypercholesterolemic fatty acids index, the more favorable the FA profile to human health.

## Milk production and composition

The decrease in dry mater intake when dairy cows were fed vegetables oil has observed in previous studies [13,64]. The decrease in dry matter intake in the present study might have been caused by the greater lipid content in the SO and LO diets (5.67 and 5.78% of ether extract in the dry mater, respectively) compared to the CC treatment (3.59% of ether extract in the dry matter). Both oils sources have a high content of UFA (84.2 and 88.4%, respectively for soybean and linseed oils), and according to [65], dairy diets containing oil sources rich in unprotected PUFA often causes a decrease in DM intake. The mechanisms behind the decreased dry matter intake are attributed to alteration of ruminal fermentation, gut motility, palatability of added fat diets, release of certain gut hormones, and oxidation of fat in the liver [65]. The reduction of nutrient intake and ruminal fermentation may reduce the rate of digestion and consequently decrease the flow of nutrients arriving to the mammary gland, which in certain cases could reduces milk production. Nevertheless, despite the reduction in dry matter intake caused by the addition of oils in the diets, this effect was not enough to affect milk yield in the present study. Perhaps because the greater content of energy contained in vegetable oils made up for the decrease in dry mater intake. This lack of difference on milk yield when animals are fed diets with high content of oil has also been observed on a recent study from our research group [14].

Inclusion of SO and LO decreased the percentage and the yield of fat in the milk of dairy cows. Compared to the CC, animals fed SO and LO produced 17.3% and 16.0% less milk fat. In contrast to our research, some other previous research did not find such a prominent decrease in the fat content of cows fed vegetable oils. Indeed, the EE % was not over the recommended by [26]. Nevertheless, it is important to note that even though the forage-concentrate ratio was 50:50, corn silage was the only forage source used on our study. According to [43], although corn silage-based diet may contain an adequate level of fiber, its effectiveness to maintain rumen function less than the fiber in grass silage-based diets, which might negatively affect normal milk fat. Additionally, the high proportion of corn grain present is this silage reduces its effectiveness of corn silage as a forage source. Indeed, vegetable oil supplements will not depress milk fat yield if roughage intake is elevated or the effectiveness of roughage fiber is sufficient to maintain normal rumen function [66,67].

The decrease in the percentage of total solids in milk on cows fed SO and LO was likely a reflection to the decreased fat production. Since total solids correspond to the summation of fat, protein, and lactose, a decrease on fat production would affect the total solid percentage directly. The decrease in fat concentration and total solids in the milk might be undesirable for certain sectors of the dairy industry. That is because milk solids is an important component for the production of certain milk derivate, like butter, cream and chesses [68]. On the other hand, the more desirable FA profile of milk fat could classify milk as "functional food", which

could make the product more profitable as consumers from developed countries are attracted to healthier foods [69,70].

## Blood metabolites

In the present study, total cholesterol, HDL, and LDL were the metabolites affected by supplementation of SO and LO. This effect was probably observed due to an increase in fatty acid intake caused by vegetable oil supplementation. On the metabolism of ruminants, the unsaturated fatty acids that leave the rumen are absorbed through the intestinal epithelium [42]. In the enterocytes the fatty acids are re-esterified to triacylglycerols and arranged mainly in chylomicrons. Through the lymphatic system they reach the bloodstream, where they will be finally directed to peripheral tissues [42]. Although total cholesterol increased in the present study, HDL levels also increased. HDL is a high-density lipoprotein, and it might work in a beneficial way by mobilizing excess cholesterol present in the blood to the liver. When it reaches the liver, excess cholesterol can be metabolized and thus prevents the accumulation of fat in peripheral tissues of cows. The lack of statistical difference between CC and vegetable oil treatments on other metabolic parameters, like AST and GGT, might mean that SO and LO can be fed in the order of 2.5% of the diet without causing negative effects on the health of dairy cows.

## Conclusion

Dietary supplementation with LO increased the concentration of C18:3 omega-3 and decreased the milk omega-6/omega-3 ratio to 2.72. The results found in the present study provide further evidence that the addition of vegetable oils in the ruminant diet makes the fatty acid profile of the milk nutritionally healthier for human diets. Future, studies should evaluate the health benefits of omega-3 and omega-6 enriched milk in the human diet.

## Acknowledgments

We are very grateful for the Faculty of Animal Science and Food Engineering for providing the animals and physical structure for the execution of this project.

## Author Contributions

**Conceptualization:** Mauricio X. S. Oliveira, Alessandra P. S. Marconi, Leriana G. Reis, Marcia S. V. Salles, Arlindo S. Netto.

**Data curation:** Camila S. R. Franco, Alessandra P. S. Marconi, Arlindo S. Netto.

**Formal analysis:** Andre S. V. Palma, Camila S. R. Franco, Arlindo S. Netto.

**Funding acquisition:** Mauricio X. S. Oliveira, Camila S. R. Franco, Alessandra P. S. Marconi, Leriana G. Reis.

**Investigation:** Mauricio X. S. Oliveira, Andre S. V. Palma, Barbara R. Reis, Camila S. R. Franco, Alessandra P. S. Marconi, Fabiana A. Shiozaki, Leriana G. Reis, Marcia S. V. Salles, Arlindo S. Netto.

**Methodology:** Mauricio X. S. Oliveira, Andre S. V. Palma, Barbara R. Reis, Camila S. R. Franco, Alessandra P. S. Marconi, Fabiana A. Shiozaki, Leriana G. Reis, Marcia S. V. Salles, Arlindo S. Netto.

**Project administration:** Mauricio X. S. Oliveira, Andre S. V. Palma, Barbara R. Reis, Camila S. R. Franco, Alessandra P. S. Marconi, Fabiana A. Shiozaki, Marcia S. V. Salles, Arlindo S. Netto.

**Resources:** Mauricio X. S. Oliveira, Marcia S. V. Salles, Arlindo S. Netto.

**Software:** Mauricio X. S. Oliveira, Camila S. R. Franco, Alessandra P. S. Marconi, Arlindo S. Netto.

**Supervision:** Mauricio X. S. Oliveira, Andre S. V. Palma, Barbara R. Reis, Fabiana A. Shiozaki, Marcia S. V. Salles, Arlindo S. Netto.

**Validation:** Mauricio X. S. Oliveira, Andre S. V. Palma, Marcia S. V. Salles, Arlindo S. Netto.

**Visualization:** Mauricio X. S. Oliveira, Andre S. V. Palma, Barbara R. Reis, Alessandra P. S. Marconi, Marcia S. V. Salles, Arlindo S. Netto.

**Writing – original draft:** Camila S. R. Franco, Alessandra P. S. Marconi, Arlindo S. Netto.

**Writing – review & editing:** Mauricio X. S. Oliveira, Andre S. V. Palma, Fabiana A. Shiozaki, Leriana G. Reis, Marcia S. V. Salles.

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
