## [Decision Letter · Decision Letter 0]

8 Oct 2020

PONE-D-20-26797

Inclusion of soybean and linseed oils in the diet of lactating dairy cows makes the milk fatty acid profile nutritionally healthier for the human diet

PLOS ONE

Dear Dr. Saran Netto,

Thank you for submitting your manuscript to PLOS ONE. After careful consideration, we feel that it has merit but does not fully meet PLOS ONE’s publication criteria as it currently stands. Therefore, we invite you to submit a revised version of the manuscript that addresses the points raised during the review process.

We look forward to receiving your revised manuscript.

Kind regards,

Juan J Loor

Academic Editor

PLOS ONE

Journal Requirements:

2.Thank you for stating the following in the Acknowledgments Section of your manuscript:

[We are grateful to the São Paulo Research Foundation (FAPESP) in São

501 Paulo, Brazil for the research funding (FAPESP Process 2015/ 19393-8) and for the

502 scholarships (FAPESP Processes 2016/23735-4 and 2016/23797-0).]

 [The funders had no role in study design, data collection and analysis, decision to publish, or preparation of the manuscript.]

3. Please amend the manuscript submission data (via Edit Submission) to include author Mauricio X. S. Oliveira.

Reviewers' comments:

Reviewer's Responses to Questions

**Comments to the Author**

1. Is the manuscript technically sound, and do the data support the conclusions?

Reviewer #1: Partly

Reviewer #2: Yes

2. Has the statistical analysis been performed appropriately and rigorously? 

Reviewer #1: I Don't Know

Reviewer #2: Yes

3. Have the authors made all data underlying the findings in their manuscript fully available?

Reviewer #1: Yes

Reviewer #2: Yes

4. Is the manuscript presented in an intelligible fashion and written in standard English?

Reviewer #1: Yes

Reviewer #2: Yes

5. Review Comments to the Author

Reviewer #1: PONE-D-20-26797 is a manuscript dealing with cows fed with two dietary oils and their effect on milk FA profile. Overall, the manuscript has some details that need to be clarified. There is a lack of care in the overall manuscript presentation.

Introduction: please write the specific biological mechanisms whereby dietary FA are shifted into healthy milk FA

Lines 96-98 Please be more specific on your hypothesis, what are the expected changes in individual FA?

Lines 103-106 re write, not sure why are you describing this, also add the reference from the pig study or refer to a code id project or something where we can visualize the overall framework

Lines 114-115 re write, you have one study divided in 10 d for adaptation and 84 d for data collection? But then in lines 120-121 you mentioned that it was 28 days for experimental period with 21 days for adaptation, please clarify

Lines 129-131 ok , but write the rationale behind it, what was the theoretical assumptions?

Lines 223-224 write the volumes per animal per collection period

Results: in tis section please add C for each individual FA

Line 311 change to Blood metabolites

LINE 328 add reference

Line 330 add reference

Lines 333-339 re write the paragraph, it seems very intricate

DISCUSSION:

Authors have discussed the observed changes superficially, please at each subsection or data set, explain the biological mechanisms behind the results. You worked with ruminants and their lipid metabolism is the key point whereby the observed changes are explain.

CONCLUSION:

Make it in 3-4 lines, explain overall result and implications of your study.

Tables: sometimes is c and sometimes is C, please use C for fatty acids nomenclature

Table 1 is it % of dry matter, please write it down

Table 2 use 2 decimals,

Table 3 use 2 decimals and explain which unit you are using

Reviewer #2: The study was done to with the objective of producing milk with fatty acid composition that is healthier for human consumption by dietary inclusion of soybean oil and linseed oil in the dairy cow diets. The results from the study confirm earlier reports that inclusion of oil in the dairy diets reduce the feed intake, decreases milk fat and alters milk fatty acid profile.

Main Questions:

1. Pl. elaborates how this study is different from the earlier studies with supplementation of different types of oils in the diet? There are numerous studies in the area that have been done using the oil supplementation. Summarise these and compare your results.

2. How did you decide on the levels of the supplemental oil in the experiment??

3. In the tables, indicate the differences between the treatments by using the different alphabets to each treatment when there are significant differences.

Minor comments:

Line 61. Change to “increase in blood….”

Line 69. Pl. be more specific. From which year and in which of the countries??

Line 105-106. Have you included any data related to evaluation of milk in the pig model? If not, remove the sentence and include the statement in the conclusion as future directions.

Line 123-130. Give brief explanation as to how the levels of oil inclusion were decided? Give suitable references.

6. PLOS authors have the option to publish the peer review history of their article (what does this mean?). If published, this will include your full peer review and any attached files.

Reviewer #1: No

Reviewer #2: No

---

## [Author Response · Author response to Decision Letter 0]

19 Nov 2020

Response to Reviewers Comments

Reviewers one and two:

AU: Thank you for your kind words and affirmation that this study is relevant and important. We greatly appreciate you taking the time to review this manuscript. Your suggestions have improved the manuscript immensely. 

Reviewer one:

Reviewer #1: PONE-D-20-26797 is a manuscript dealing with cows fed with two dietary oils and their effect on milk FA profile. Overall, the manuscript has some details that need to be clarified. There is a lack of care in the overall manuscript presentation.

AU: The authors agree that there were some details missing in the first submission. We have worked on your detailed remarks and we believe you will find it clarifying. In the first submission, we focused our discussion on the nutritional quality of milk on a human perspective. Nevertheless, the authors agree that extending the discussion section by explaining more on the ruminant lipid metabolism is beneficial to the manuscript. Hopefully you will find this second version more tidily. Please see more details bellow.

Introduction: please write the specific biological mechanisms whereby dietary FA are shifted into healthy milk FA

AU: We have added a new section explaining this specific mechanism. Please find this newly added information on lines 91-101.

Lines 96-98 Please be more specific on your hypothesis, what are the expected changes in individual FA?

AU: We have clearly stated that on our hypothesis and this change is found on lines 105-109.

Lines 103-106 re write, not sure why are you describing this, also add the reference from the pig study or refer to a code id project or something where we can visualize the overall framework

AU: We have now realized the mentioning the second study is causing much confusion on the reader’s perspective. Hence, we have decided to follow reviewer #2 advice and remove the mentions from the swine model on the body of the text and simply include a short statement in the conclusion as future directions. Because of that, we will not be adding a reference to the study. Finally, the swine manuscript is currently under review. Unfortunately, until the manuscript is published, you will only be able to find the Mater’s thesis published on the native language of where the study was conducted (https://www.teses.usp.br/teses/disponiveis/74/74131/tde-07052019-075412/publico/ME9691119COR.pdf). The abstract section is written in the English. 

Lines 114-115 re write, you have one study divided in 10 d for adaptation and 84 d for data collection? But then in lines 120-121 you mentioned that it was 28 days for experimental period with 21 days for adaptation, please clarify

AU: We realize that this is confusing and could have been better explained: the very first 10 days of study was designated to adapt the animals to the newly modified environment, which consisted of the stalls, divided by electric fences, in which they were kept during the whole study. At this time, all animals were fed the same diet. After the first ten days, each animal received its designated experimental diet from the Latin square design. There were three treatment diets and hence, three experimental periods consisting of 28 days each, which equals a total experimental period of 84 days. The first 21 days were used to adapt the animals to the new diet, and the last seven days of each experimental period were used to data collection. You will find adjustments to the manuscript from line 123 to line 129.

Lines 129-131 ok , but write the rationale behind it, what was the theoretical assumptions?

AU: See rationale on lines 141-142.

Lines 223-224 write the volumes per animal per collection period

AU: The volumes were added to the main text and you can see the details on line 238 to 240. 

Results: in tis section please add C for each individual FA

AU: We have fixed that issue in the Results section (line 280 to 357) and throughout the manuscript.

Line 311 change to Blood metabolites

AU: Agreed. See changes on line 342 and 521.

LINE 328 add reference

AU: Reference has been added and can be find on line 361.

Line 330 add reference

AU: Reference has been added and can be find on line 363.

Lines 333-339 re write the paragraph, it seems very intricate

AU: As we explained before, we realize that mentioning the swine model in this paper is causing confusion on readers. Because of that, the authors have decided to follow the instruction from reviewer two and remove the mentions from the swine model on the body of the text and simply include a statement in the conclusion as future directions.

DISCUSSION:

Authors have discussed the observed changes superficially, please at each subsection or data set, explain the biological mechanisms behind the results. You worked with ruminants and their lipid metabolism is the key point whereby the observed changes are explain.

AU: This is a very good suggestion. Because our ultimate goal is to evaluate if the milk produced in this study is beneficial to other species (swine and eventually humans), we focused our discussion on the nutritional quality of milk on a human perspective. Nevertheless, the authors agree that extending the discussion section by explaining more on the ruminant lipid metabolism is beneficial to the manuscript. Hence, in the present version, you will find a more detailed discussion on the de novo synthesis of fatty acids in the mammary gland (lines 370 to 386), the effect of unsaturated fatty acid supplementation on ruminal microbes (lines 395 to 408), ruminal biohydrogenation (lines 409 to 417), overall lipid metabolism in ruminants (lines 425 to 435), and lipid supplementation and its effect of dry matter intake (lines 480 to 496). We realize that these discussions were missing in the first submission and agree that they have enhanced the quality of our manuscript.

CONCLUSION:

Make it in 3-4 lines, explain overall result and implications of your study.

AU: We have shortened the study conclusion. You can see changes starting in line 538. We have decided to follow reviewer #2 advice and remove the mentions from the swine model on the body of the text and simply include a short statement in the conclusion as future directions. Because of that, our conclusion is a little over the suggested 3-4 lines.

Tables: sometimes is c and sometimes is C, please use C for fatty acids nomenclature

AU: This has been fixed. You can see the changes starting on lines 222 and 290.

Table 1 is it % of dry matter, please write it down

AU: That is correct – the values are expressed in dry matter percentage. We have rewritten the title of table one and the changes can be found on line 146 and 147.

Table 2 use 2 decimals,

AU: The adjustments have been made. You will find the changes on line 222 to 224.

Table 3 use 2 decimals and explain which unit you are using

AU: The adjustments have been made. You will find the changes starting on lines 289 to 292.

Reviewer two

Reviewer #2: The study was done to with the objective of producing milk with fatty acid composition that is healthier for human consumption by dietary inclusion of soybean oil and linseed oil in the dairy cow diets. The results from the study confirm earlier reports that inclusion of oil in the dairy diets reduce the feed intake, decreases milk fat and alters milk fatty acid profile.

AU: Thank you so much for taking the time to review our manuscript. Please see comments for specific remarks below.

Main Questions:

1. Pl. elaborates how this study is different from the earlier studies with supplementation of different types of oils in the diet? There are numerous studies in the area that have been done using the oil supplementation. Summarise these and compare your results.

AU: This is a good question. Indeed, the present study was performed to provide additional evidence of the effects of oil supplementation on the fat fraction of milk from dairy cows. We realize that the literature on oil supplementation is not scarce worldwide; however, there are much fewer publications in the literature describing the results of trials that evaluated oils under tropical conditions. Moreover, unlike other studies, the present research was carried out concomitantly with another animal nutrition trial. We evaluated the effects of feeding the milk produced in the present study on the health and reproduction of female Sus scrofa domesticus and their offspring. Once we have this study and the swine study, our next step is to compare the omega-3, omega-6 and the control milk in the human diet. It is our understanding that very few studies in the available literature have evaluated the effects of naturally enhanced milk on health parameters of other species. On the other hand, we realize that mentioning the swine model in this paper is causing confusion on readers. Because of that, the authors have decided to follow your instruction and remove the mentions from the swine model on the body of the text and simply include a statement in the conclusion as future directions.

Finally, in this new version, you will find a more detailed discussion on the results from the present study and previous studies (lines 370 to 386; lines 395 to 408; lines 409 to 417; 480 to 496, and finally, 499 to 510).

2. How did you decide on the levels of the supplemental oil in the experiment??

AU: The 2.5% level of inclusion of oil in the total dry matter was decided based on previous studies from our research group. In the past, we have evaluated both 3% and 6% (Welter et al., 2016) our 4% (Salles et al., 2019) of vegetable oils in the DM of dairy cows. We have observed a decrease in dry mater intake and subsequent decrease in milk yield (Welter et al., 2016) when using 3 or 6% of oil and also a decrease on the protein content of milk (Salles et al., 2019). 

Finally, findings from a different research group (Benchaar et al., 2012) suggested that including up to 2.5% of linseed oil could be safely supplemented to forage-based diets of dairy cows to enrich milk with potential health-beneficial fatty acid with little detrimental effect on digestion, rumen function, and animal performance. 

3. In the tables, indicate the differences between the treatments by using the different alphabets to each treatment when there are significant differences.

AU: The authors have taken your suggestion into consideration and carefully discussed the possibility of using letters to indicate statistical difference between treatments. However, it is our understanding that using different letters whenever a treatment is significantly different is what is done under a multiple comparison statistical analysis. On the other hand, considering that on our data we performed orthogonal contrasts (OC), we believe that simply adding alphabets letters to each treatment means would not be appropriate. 

Our treatment groups are structured (NO oil, PRESENCE of soybean oil, PRESENCE of linseed oil), and we believe the OC evaluation is the most appropriate. Our structure is based on the evaluation of the “presence of oil in the diet” (contrast one) and “the two oil types” (contrast two). The OC analysis not only informs us about the differences in treatments (like a multiple comparison analysis would), but it has hr advantage of allowing us to explore a whole relationship that would not otherwise be addressable. 

If you believe that OC is not the appropriate way to go here, please let us know and we will reconsider changing the OC analysis to a multiple comparison test.

Minor comments:

Line 61. Change to “increase in blood….”

AU: We have corrected that. See changes on line 58.

Line 69. Pl. be more specific. From which year and in which of the countries??

AU: We have modified that statement and provided suitable reference. You can find that on lines 66 to 68.

Line 105-106. Have you included any data related to evaluation of milk in the pig model? If not, remove the sentence and include the statement in the conclusion as future directions.

AU: We realize that mentioning the pig model in this paper is causing confusion on readers. Because of that, the authors have decided to follow the your instruction and remove the mentions from the swine model on the body of the text and simply include a statement in the conclusion as future directions

Line 123-130. Give brief explanation as to how the levels of oil inclusion were decided? Give suitable references.

We have added that information with proper reference to lines 135 to 137.

---

## [Decision Letter · Decision Letter 1]

18 Jan 2021

Inclusion of soybean and linseed oils in the diet of lactating dairy cows makes the milk fatty acid profile nutritionally healthier for the human diet

PONE-D-20-26797R1

Dear Dr. Saran Netto,

We’re pleased to inform you that your manuscript has been judged scientifically suitable for publication and will be formally accepted for publication once it meets all outstanding technical requirements.

Kind regards,

Juan J Loor

Academic Editor

PLOS ONE

Additional Editor Comments (optional):

Reviewers' comments:

Reviewer's Responses to Questions

**Comments to the Author**

1. If the authors have adequately addressed your comments raised in a previous round of review and you feel that this manuscript is now acceptable for publication, you may indicate that here to bypass the “Comments to the Author” section, enter your conflict of interest statement in the “Confidential to Editor” section, and submit your "Accept" recommendation.

Reviewer #1: All comments have been addressed

2. Is the manuscript technically sound, and do the data support the conclusions?

Reviewer #1: Yes

3. Has the statistical analysis been performed appropriately and rigorously? 

Reviewer #1: I Don't Know

4. Have the authors made all data underlying the findings in their manuscript fully available?

Reviewer #1: Yes

5. Is the manuscript presented in an intelligible fashion and written in standard English?

Reviewer #1: Yes

6. Review Comments to the Author

Reviewer #1: Authors have done changes accordingly

7. PLOS authors have the option to publish the peer review history of their article (what does this mean?). If published, this will include your full peer review and any attached files.

Reviewer #1: **Yes: **Einar Vargas-Bello-Pérez

---

## [Editor Report · Acceptance letter]

25 Jan 2021

PONE-D-20-26797R1 

Inclusion of soybean and linseed oils in the diet of lactating dairy cows makes the milk fatty acid profile nutritionally healthier for the human diet 

Dear Dr. Netto:

I'm pleased to inform you that your manuscript has been deemed suitable for publication in PLOS ONE. Congratulations! Your manuscript is now with our production department. 

Kind regards, 

on behalf of

Dr. Juan J Loor 

Academic Editor

PLOS ONE